# Near-Optimal Multi-Agent Learning for Safe Coverage Control

**Manish Prajapat**
ETH Zurich
manishp@ai.ethz.ch

**Matteo Turchetta**
ETH Zurich
matteotu@inf.ethz.ch

**Melanie N. Zeilinger**[†]
ETH Zurich
mzeilinger@ethz.ch

**Andreas Krause**[†]
ETH Zurich
krausea@ethz.ch

## Abstract

In multi-agent coverage control problems, agents navigate their environment to reach locations that maximize the coverage of some density. In practice, the density is rarely known *a priori*, further complicating the original NP-hard problem. Moreover, in many applications, agents cannot visit arbitrary locations due to *a priori* unknown safety constraints. In this paper, we aim to efficiently learn the density to approximately solve the coverage problem while preserving the agents' safety. We first propose a conditionally linear submodular coverage function that facilitates theoretical analysis. Utilizing this structure, we develop MACOPT, a novel algorithm that efficiently trades off the exploration-exploitation dilemma due to partial observability, and show that it achieves sublinear regret. Next, we extend results on single-agent safe exploration to our multi-agent setting and propose SAFEMAC for safe coverage and exploration. We analyze SAFEMAC and give first of its kind results: near optimal coverage in finite time while provably guaranteeing safety. We extensively evaluate our algorithms on synthetic and real problems, including a biodiversity monitoring task under safety constraints, where SAFEMAC outperforms competing methods.

## 1   Introduction

In multi-agent coverage control (MAC) problems, multiple agents coordinate to maximize coverage over some spatially distributed events. Their applications abound, from collaborative mapping [1], environmental monitoring [2], inspection robotics [3] to sensor networks [4]. In addition, the coverage formulation can address core challenges in cooperative multi-agent RL [5, 6], e.g., *exploration* [7], by providing high-level goals. In these applications, agents often encounter safety constraints that may lead to critical accidents when ignored, e.g., obstacles [8] or extreme weather conditions [9, 10].

Deploying coverage control solutions in the real world presents many challenges: (*i*) for a given density of relevant events, this is an *NP hard problem* [11]; (*ii*) such *density* is *rarely known* in practice [2] and must be learned from data, which presents a complex active learning problem as the quantity we measure (the density) differs from the one we want to optimize (its coverage); (*iii*) agents often operate under *safety-critical* conditions, [8–10], that may be *unknown a priori*. This requires cautious exploration of the environment to prevent catastrophic outcomes. While prior work addresses subsets of these challenges (see Section 7), we are not aware of methods that address them jointly.

---

† Joint supervision. Code available at https://github.com/manish-pra/SafeMaC

36th Conference on Neural Information Processing Systems (NeurIPS 2022).

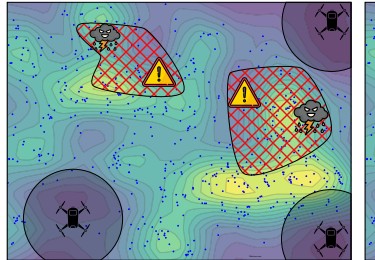 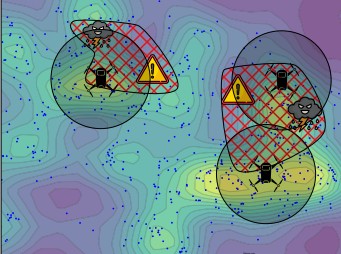 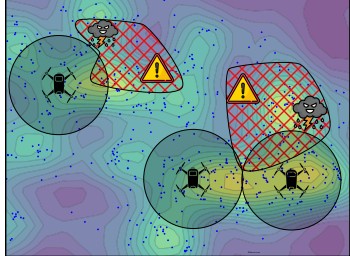

(a) Safe initialization        (b) Coverage maximization        (c) Safe coverage maximization

Figure 1: The three drones aim to maximize the gorilla nests' coverage (grey shaded circle) while avoiding unsafe extreme weather zones (red cross pattern). The contours (yellow is high and purple is low) represent the density of gorilla nests (blue dots). The density and the constraint are a-prior unknown. a) To be safe, drones apply a conservative strategy and do not explore, which results in poor coverage. In b), the drones maximize coverage but get destroyed in extreme weather. c) shows SAFEMAC solution. The drones strike a balance, trading off between learning the density and the constraints, and thus achieve near-optimal coverage while always being safe.

This work makes the following contributions toward efficiently solving safe coverage control with *a-priori* unknown objectives and constraints. **Firstly**, we model this multi-agent learning task as a *conditionally linear* coverage function. We use the *monotonocity* and the *submodularity* of this function to propose MACOPT, a new algorithm for the unconstrained setting that enjoys sublinear cumulative regret and efficiently recommends a near-optimal solution. **Secondly**, we extend GOOSE [12], an algorithm for single agent safe exploration, to the multi-agent case. Combining our extension of GOOSE with MACOPT, we propose SAFEMAC, a novel algorithm for safe multi-agent coverage control. We analyze it and show it attains a near-optimal solution in a finite time. **Finally**, we demonstrate our algorithms on a synthetic and two real world applications: safe biodiversity monitoring and obstacle avoidance. We show SAFEMAC finds better solutions than algorithms that do not actively explore the feasible region and is more sample efficient than competing near-optimal safe algorithms.

## 2 Problem Statement

We present the safety-constrained multi-agent coverage control problem (Fig. 1) that we aim to solve.

**Coverage control**. Coverage control models situations where we want deploy a swarm of dynamic agents to maximize the coverage of a quantity of interest, see Fig. 1. Formally, given a finite[1] set of possible locations $V$, the goal of coverage control is to maximize a function $F : 2^V \to \mathbb{R}$ that assigns to each subset, $X \subseteq V$, the corresponding coverage value. For $N$ agents, the resulting problem is $\arg\max_{X : |X| \leq N} F(X)$. The discrete domain $V$ can be represented by a graph, where nodes represent locations in the domain, and an edge connects node $v$ to $v'$ if the agent can go from $v$ to $v'$. This corresponds to a deterministic MDP where locations are states and edges represent transitions.

**Sensing region**. Depending on the application, we may use different definitions of $F$. Here, we model cases where agent $i$ at location $x^i$ covers a limited sensing region around it, $D^i$. While $D^i$ can be any connected subset of $V$, in practice it is often a ball centered at $x^i$. Given a function $\rho : V \to \mathbb{R}$ denoting the density of a quantity of interest at each $v \in V$, our coverage objective is

$$F(X; \rho, V) = \sum_{x^i \in X} \sum_{v \in D^{i-}} \rho(v)/|V|, \tag{1}$$

where $D^{i-} := D^i \setminus D^{1:\, i-1}$ indicates the elements in $V$ covered by agent $i$ but not agents $1 : i - 1$, $D^{1:\, i-1} = \cup_{j=1}^{i-1} D^j$ and $|V|$ denotes cardinality of the domain $V$.

**Safety**. In many real-world problems, agents cannot go to arbitrary locations due to safety concerns. To model this, we introduce a constraint function $q : V \to \mathbb{R}$ and we consider safe all locations $v$ satisfying $q(v) \geq 0$. Such constraint restricts the space of possible solutions of our problem in two ways. First, it prevents agents from monitoring from unsafe locations. Second, depending on its dynamics, agent $i$ may be unable to safely reach a disconnected safe area starting from $x_0^i$, see

---

[1]Continuous domains can be handled via discretization

Appendix A.3. We denote with $\bar{R}_{\epsilon_q}(\{x_0^i\})$ the largest safely reachable region starting from $x_0^i$ and with $\mathcal{B}$ a collection of batches of agents such that all agents in the same batch $B$ share the same safely reachable set, $\forall i, j \in B \colon \bar{R}_{\epsilon_q}(\{x_0^i\}) \cap \bar{R}_{\epsilon_q}(\{x_0^j\}) \neq \emptyset$, see Appendix A for formal definitions. Based on this, we define the safely reachable control problem

$$\sum_{B \in \mathcal{B}} \max_{X^B \in \bar{R}_{\epsilon_q}(X_0^B)} F(X^B; \rho, \bar{R}_{\epsilon_q}(X_0^B)), \tag{2}$$

where $X_0^B = \{x_0^i\}_{i \in B}$ are the starting locations of all agents in $B$ and $\bar{R}_{\epsilon_q}(X_0^B) = \bigcup_{i \in B} \bar{R}_{\epsilon_q}(\{x_0^i\})$ indicates the largest safely reachable region from any point $x_0^i$ for all $i$ in $B$ (since the agents have the same dynamics, $\bar{R}_{\epsilon_q}(X_0^B) = \bar{R}_{\epsilon_q}(\{x_0^i\}), \forall i \in B$). In safety-critical monitoring, there may be unreachable safe regions. However, since agents should be able to collect measurements if required, we focus only on covering the safely reachable region.

**Unknown density and constraint**. In practice, the density $\rho$ and the constraint $q$ are often unknown *a priori*. However, the agents can iteratively obtain noisy measurements of their values at target locations. We consider synchronous measurements, i.e., we wait until all agents have collected the desired measurement for the current iteration before moving to the next one. Here, we focus on the high-level problem of choosing informative locations, rather than the design of low-level motion planning [2]. Therefore, our goal is to find an approximate solution to the problem in Eq. (2) preserving safety throughout exploration, i.e., at every location visited by the agents, while taking as few measurements as possible in case the dynamics of the agents are deterministic and known as in [12].

## 3 Background

This section presents foundational ideas that our method builds on. In particular, it discusses (*i*) monotone submodular functions and (*ii*) previous work on single-agent safe exploration.

**Submodularity**. Optimizing a function defined over the power set of a finite domain, $V$, scales combinatorially with the size of $V$ in general. In special cases, we can exploit the structure of the objective to find approximate solutions efficiently. Monotone submodular functions are one example of this.

A set function $F : 2^V \to \mathbb{R}$ is *monotone* if for all $A \subseteq B \subset V$ we have $F(A) \leq F(B)$. It is *submodular* if $\forall A \subseteq B \subseteq V, v \in V \setminus B$, we have, $F(A \cup \{v\}) - F(A) \geq F(B \cup \{v\}) - F(B)$. In coverage control, this means adding $v$ to $A$ yields at least as much increase in coverage than adding $v$ to $B$, if $A \subseteq B$. Crucially, [13] guarantees that the greedy algorithm produces a solution within a factor of $(1 - 1/e)$ of the optimal solution for problems of the type $\arg\max_{X:|X| \leq N} F(X; \rho, V)$, when $F$ is monotone and submodular. In practice, the greedy algorithm often outperforms this worst-case guarantee [14] and guaranteeing a solution better than $(1 - 1/e)$ factor is NP hard [15].

The coverage function in Eq. (1) is a conditionally linear, monotone and submodular function (proof in Appendix B), which lets us use the results above to design our algorithm for safe coverage control.

**Goal-oriented safe exploration**. GoOSE [12] is a single-agent safe exploration algorithm that extends unconstrained methods to safety-critical cases. Concretely, it maintains under- and over-approximations of the feasible set, called pessimistic and optimistic safe sets. It preserves safety by restricting the agent to the pessimistic safe set. It efficiently explores the objective by letting the original unconstrained algorithm recommend locations within the optimistic safe set. If such recommendations are provably safe, the agent evaluates the objective there. Otherwise, it evaluates the constraint at a sequence of safe locations to prove that such recommendation is either safe, which allows it to evaluate the objective, or unsafe, which triggers the unconstrained algorithm to provide a new recommendation.

**Assumptions**. To guarantee safety, GoOSE makes two main assumptions. First, it assumes there is an initial set of safe locations, $X_0$, from where the agent can start exploring. Second, it assumes the constraint is sufficiently well-behaved, so that we can use data to infer the safety of unvisited locations. Formally, it assumes the domain $V$ is endowed with a positive definite kernel $k^q(\cdot, \cdot)$, and that the constraint's norm in the associated *Reproducing Kernel Hilbert Space* [16] is bounded, $\|q\|_{k^q} \leq B_q$. This lets us use Gaussian Processes (GPs) [17] to construct high-probability confidence intervals for $q$. We specify the GP prior over $q$ through a mean function, which we assume to be

---

[2] Agents can use their transition graph to find a path between two goals. In a continuous domain, the path can be tracked with a controller (e.g., MPC)

zero everywhere w.l.o.g., $\mu(v) = 0, \forall v \in V$, and a kernel function, $k$, that captures the covariance between different locations. If we have access to $T$ measurements, at $V_T = \{v_t\}_{t=1}^T$ perturbed by i.i.d. Gaussian noise, $y_T = \{q(v_t) + \eta_t\}_{t=1}^T$ with $\eta_t \sim \mathcal{N}(0, \sigma^2)$, we can compute the posterior mean and covariance over the constraint at unseen locations $v, v'$ as $\mu_T(v) = k_T^\top(v)(K_T + \sigma^2 I)^{-1} y_T$ and $k_t(v, v') = k(v, v') - k_T^\top(v)(K_T + \sigma^2 I)^{-1} k_T(v')$, where $k_T(v) = [k(v_1, v), ..., k(v_T, v)]^\top$, $K_T$ is the positive definite kernel matrix $[k(v, v')]_{v,v' \in V_T}$ and $I \in \mathbb{R}^{T \times T}$ denotes the identity matrix.

In this work, we make the same assumptions about the safe seed and the regularity of $q$ and $\rho$.

**Approximations of the feasible set**. Based on the GP posterior above, GOOSE builds monotonic confidence intervals for the constraint at each iteration $t$ as $l_t^q(v) := \max\{l_{t-1}^q(v), \mu_{t-1}^q(v) - \beta_t^q \sigma_{t-1}^q(v)\}$ and $u_t^q(v) := \min\{u_{t-1}^q(v), \mu_{t-1}^q(v) + \beta_t^q \sigma_{t-1}^q(v)\}$, which contain the true constraint function for every $v \in V$ and $t \geq 1$, with high probability if $\beta_t^q$ is selected as in [18] or Section 5. GOOSE uses these confidence intervals within a set $S \subseteq V$ together with the $L_q$-Lipschitz continuity of $q$, to define operators that determine which locations are safe in plausible worst- and best-case scenarios,

$$p_t(S) = \{v \in V, |\exists z \in S : l_t^q(z) - L_q d(v, z) \geq 0\}, \tag{3}$$
$$o_t^{\epsilon_q}(S) = \{v \in V, |\exists z \in S : u_t^q(z) - \epsilon_q - L_q d(v, z) \geq 0\}. \tag{4}$$

Notice that the pessimistic operator relies on the lower bound, $l^q$, while the optimistic one on the upper bound, $u^q$. Moreover, the optimistic one uses a margin $\epsilon_q$ to exclude "barely" safe locations as the agent might get stuck learning about them. Finally, to disregard locations the agent could not safely reach or from where it could not safely return, GOOSE introduces the $R^{\text{ergodic}}(\cdot, \cdot)$ operator. $R^{\text{ergodic}}(p_t(S), S)$ indicates locations in $S$ or locations in $p_t(S)$ reachable from $S$ and from where the agent can return to $S$ along a path contained in $p_t(S)$. Combining $p_t(S)$ and $R^{\text{ergodic}}(\cdot, \cdot)$, GOOSE defines the pessimistic and ergodic operator $\tilde{P}_t(\cdot)$, which it uses to update the pessimistic safe set. Similarly, it defines $\tilde{O}_t(\cdot)$ using $o_t^{\epsilon_q}(\cdot)$ to compute the optimistic safe set.

# 4 MACOPT and SAFEMAC

This section presents MACOPT and SAFEMAC, our algorithms for unconstrained and safety-constrained multi-agent coverage control, which we then formally analyze in Section 5.

## 4.1 MACOPT: unconstrained multi-agent coverage control

**Greedy sensing regions**. In sequential optimization, it is crucial to balance exploration and exploitation. GP-UCB [19] is a theoretically sound strategy to strike such a trade-off that works well in practice. Agents evaluate the objective at locations that maximize an upper confidence bound over the objective given by the GP model such that locations with either a high posterior mean (exploitation) or standard deviation (exploration) are visited. We construct a valid upper confidence bound for the coverage $F(X)$ starting from our confidence intervals on $\rho$, by replacing the true density $\rho$ with its upper bound $u_t^\rho$ in Eq. (1). Next, we apply the greedy algorithm to this upper bound (Line 3 of Algorithm 1) to select $N$ candidate locations for evaluating the density. However, this simple exploration strategy may perform poorly, due to the fact that in order to reduce the uncertainty over the coverage $F$ at $X$, we must learn the density $\rho$ at all locations inside the sensing region, $\bigcup_{i=1}^N D^i$, rather than simply at $X$. It is a form of partial monitoring [20], where the objective $F$ differs from the quantity we measure, i.e., the density $\rho$. Next, we explain how to choose locations where to observe the density for a given $X$.

**Uncertainty sampling**. Given location assignments $X$ for the agents, we measure the density to efficiently learn the function $F(X)$. Intuitively, agent $i$ observes the density where it's most uncertain within the area it covers that is not covered by agents $\{1, \ldots i-1\}$, i.e., $D_t^{i^-}$ (Line 4 of Alg. 2, Fig. 2a).

**Stopping criterion**. The algorithm terminates when a near-optimal solution is achieved. Intuitively, this occurs when the uncertainty about the coverage value of the greedy recommendation is low. Formally, we require the sum of the uncertainties over the sampling targets to be below a threshold, i.e. , $w_t = \sum_{i=1}^N u_{t-1}^\rho(x_t^{g,i}) - l_{t-1}^\rho(x_t^{g,i}) \leq \epsilon_\rho$ (Line 3 of Algorithm 2). Importantly, this stopping criterion requires the confidence intervals to shrink only at regions that potentially maximize the coverage.

**MACOPT**. Now, we introduce MACOPT in Algorithm 2. At round $t$, we select the sensing locations for the agents, $X_t$, by greedily optimizing the upper confidence bound of the coverage. Then, each

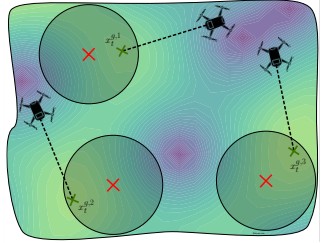

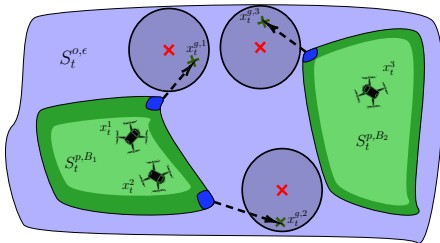

(a) Uncertainty sampling in MACOPT       (b) Illustration of multi-agent GOOSE

Figure 2: a) The contours represent the density uncertainty, and the red $\times$'s correspond to the maximum coverage locations evaluated by the GREEDY Algorithm 1. While these locations maximize coverage, they may not be informative about the coverage since the uncertainty can be low. Therefore, the agents collect measurements at the maximum uncertainty of the density in a disc (green $\times$'s, $x_t^{g,i}$), also known as uncertainty sampling. b) In a constrained environment, SAFEMAC evaluates $x_t^{g,i}$ for all agents in the optimistic set $S_t^{o,\epsilon_q}$ (violet) and set it as a next goal. It forms an expander region (dark blue) to safely expand the pessimistic safe set $S_t^p$ (green) toward the goal.

agent $i$ collects noisy density measurements at the points of highest uncertainty within $D_t^{i-}$. Finally, we update our GP over the density and, if the sum of maximum uncertainties within each sensing region is small, we stop the algorithm.

## 4.2   SAFEMAC: safety-constrained multi-agent coverage control

**Intuition**. We adopt a perspective similar to GOOSE as we separate the exploration of the safe set from the maximization of the coverage. Given an over and under approximation of the safe set (whose computation is discussed later), we want to explore optimistically optimal goals for each agent, similar to MACOPT. To this end, we find the maximizers of the density upper bound in the optimistic safe set with the GREEDY algorithm. Then, we define sampling goals to learn the coverage at those locations.

**Phases of SAFEMAC**. Coverage values depend both on the density and the feasible region (Eq. (2)). Thus, there are two sensible sampling goals given a disk assignment: i) *optimistic coverage*: if we are uncertain about the density within the disks, we target locations with the highest density uncertainty (Line 6 of Algorithm 4); ii) *optimistic exploration*: if we know the density within the disk but there are locations under it that we cannot classify as either safe (in $S^p$) or unsafe (in $V \setminus S^{o,\epsilon_q}$), we target those with the highest constraint uncertainty among them (Line 8). If all the goal locations are safe with high probability, which can only happen during *optimistic coverage*, we safely evaluate the density there (Line 19). Otherwise, we explore the constraint with a goal directed strategy that aims at classifying them as either safe or unsafe similar to GOOSE (Line 9-12). In case this changes the topological connection of the optimistic feasible set, we recompute the disks as this may change GREEDY's output (Line 15-17). We repeat this loop until we know the feasibility of all the points under the disks recommended by GREEDY and their density uncertainty is low (Line 4). Next, we explain how the multiple agents coordinate their individual safe regions to evaluate a goal (MACOPT *in batches*), how the agents progress toward their goals (*safe expansion*) and finally we describe SAFEMAC *convergence*.

**MACOPT in batches**. In the multi-agent setting of GOOSE (see Fig. 2b), each agent $i$ maintains $S_t^{p,i}$ a pessimistic (or $S_t^{o,\epsilon_q,i}$ an optimistic) belief of the safe locations, obtained by iteratively applying $\tilde{P}_t(\cdot)$ the pessimistic ( or $\tilde{O}_t(\cdot)$ the optimistic) ergodic operators (see Section 3) to the previous pessimistic belief $S_{t-1}^{p,i}$ (Line 11 of Algorithm 4). Since the agents cannot navigate to an arbitrary location in the constrained case, SAFEMAC computes coverage maximizers on a restricted region, obtained by ignoring the known unsafe locations. To denote such a restricted region, we define a union set $S_t^{u,i} :=$ $S_t^{o,\epsilon_q,i} \cup S_t^{p,i}$, which is the largest set known to be optimistically or pessimistically safe up to time $t$. Moreover, if the agents are topologically disconnected, they cannot travel from one safe region to another and the best strategy for any batch of agents is to maximize coverage locally. For this, we form a collection of batches $\mathcal{B}_t$, such that any batch $B \in \mathcal{B}_t$ contains agents that lie in topologically connected regions determined by the union set (Line 13-14 ). SAFEMAC computes a GREEDY solution for each $B \in \mathcal{B}_t$ in their corresponding $S_t^{u,B} := \cup_{i \in B} S_t^{u,i}$. This is the largest set where the agents can find an

**Algorithm 1** Greedy UCB (GREEDY)

1: **Inputs** $u^\rho_{t-1}, l^\rho_{t-1}, B, S^u_t$
2: **for** $i = 1, 2, ..., |B|$ **do**
3:     $x^i_t \leftarrow \arg\max_{x^i} \sum_{v \in D^i \setminus D^{1:i-1}_t \cap S^u_t} u^\rho_{t-1}(v)$
4:     $x^{g,i}_t \leftarrow \arg\max_{v \in D^i \setminus D^{1:i-1}_t \cap S^u_t} u^\rho_{t-1}(v) - l^\rho_{t-1}(v)$
5: $w_t \leftarrow \sum_{i=1}^{|B|} u^\rho_{t-1}(x^{g,i}_t) - l^\rho_{t-1}(x^{g,i}_t)$
6: **Return** $X^B_t, w_t$

---

**Algorithm 2** MACOPT

1: **Inputs** $X_0, \epsilon_\rho, V, GP_\rho, t \leftarrow 1$
2: $X_1, w_1 \leftarrow$ GREEDY$(u^\rho_0, l^\rho_0, [N], V)$
3: **while** $w_t > \epsilon_\rho$ **do**
4:     $\forall i, x^{g,i}_t \leftarrow \arg\max_{v \in D^{i-}_t} u^\rho_{t-1}(v) - l^\rho_{t-1}(v)$
5:     $\forall i, y^i_{\rho_t} = \rho(x^{g,i}_t) + \eta_\rho$, Update GP
6:     $t \leftarrow t + 1$
7:     $X_t, w_t \leftarrow$ GREEDY$(u^\rho_{t-1}, l^\rho_{t-1}, [N], V)$
8: **Recommend** $X_t$

---

**Algorithm 3** Safe Expansion (SE)

1: **Inputs** $S^{o,\epsilon_q}_t, S^p_t, x^g_t$
2: $A_t(p) \leftarrow \{v \in S^{b,\epsilon_q}_t \setminus \tilde{p}_t(S^p_t) | h(v) = p\}$
3: $W^{\epsilon_q}_t \leftarrow \{v \in S^p_t | u^q_t(v) - l^q_t(v) > \epsilon_q\}$
4: $\alpha^\star \leftarrow \max \alpha$ s.t. $|G^{\epsilon_q}_t(\alpha)| > 0$
5: **if** Optimization problem feasible **then**
6:     $v_t \leftarrow \arg\max_{v \in G^{\epsilon_q}_t(\alpha^\star)} u^q_t(v) - l^q_t(v)$
7:     Update GP with $y_t = q(v_t) + \eta_q$

---

**Algorithm 4** SAFEMAC

1: **Inputs** $X_0, L_q, \epsilon_\rho, V, GP_\rho, GP_q$
2: $\forall i, S^{p,i}_0 \leftarrow X_0, S^{o,\epsilon_q,i}_0 \leftarrow V, t \leftarrow 1$
3: $X_1, w_1 \leftarrow$ GREEDY$(u^\rho_0, l^\rho_0, [N], V)$
4: **while** $\forall i, (S^{o,\epsilon_q,i}_{t-1} \setminus S^{p,i}_{t-1}) \cap D^i_t \neq \emptyset$ or $w_t > \epsilon_\rho$ **do**
5:     **if** $w_t > \epsilon_\rho$ **then**
6:        $\forall i, x^{g,i}_t \leftarrow \arg\max_{v \in D^{i-}_t} u^\rho_{t-1}(v) - l^\rho_{t-1}(v)$
7:     **else**
8:        $\forall i, x^{g,i}_t \leftarrow \arg\max_{v \in (S^{o,\epsilon_q,i}_{t-1} \setminus S^{p,i}_{t-1}) \cap D^i_t} u^q_{t-1}(v) - l^q_{t-1}(v)$
9:     **if** $\exists i \in [N], x^{g,i}_t \notin S^{p,i}_t$ **then**
10:       SE$(S^{o,\epsilon_q,i}_{t-1}, S^{p,i}_{t-1}, x^{g,i}_t), \forall i : x^{g,i}_t \notin S^{p,i}_t$
11:       $S^{p,i}_t \leftarrow \tilde{P}_t(S^{p,i}_{t-1}), S^{o,\epsilon_q,i}_t \leftarrow \tilde{O}^{\epsilon_q}_t(S^{p,i}_{t-1}), \forall i$
12:       $t \leftarrow t + 1$
13:     $\forall i, \mathcal{B}'_t(i) = \{j \in [N] | S^{u,i}_t \cap S^{u,j}_t \neq \emptyset\}$
14:     $\mathcal{B}_t = \bigcup_{i \in [N]} \mathcal{B}'_t(i)$
15:     **if** *for any* $B \in \mathcal{B}_t, S^{u,B}_t \neq S^{u,B}_{t-1}$ **then**
16:       $X_t, w_t \leftarrow$ GREEDY$(u^\rho_{t-1}, l^\rho_{t-1}, B, S^{u,B}_t)$
17:       $\forall i, x^{g,i}_t \leftarrow \arg\max_{v \in D^{i-}_t} u^\rho_{t-1}(v) - l^\rho_{t-1}(v)$
18:     **if** $\forall i, x^{g,i}_t \in S^{p,i}_t$ and $w_t > \epsilon_\rho$ **then**
19:       $\forall i, y^i_{\rho_t} = \rho(x^{g,i}_t) + \eta_\rho$
20:       Update GP i.e, compute $u^\rho_t, l^\rho_t$
21:       $t \leftarrow t + 1$
22:       $X_t, w_t \leftarrow$ GREEDY$(u^\rho_{t-1}, l^\rho_{t-1}, B, S^{u,B}_{t-1})$
23: **Recommend** $X_t$

---

optimistically safe path to travel. Analogous to $\mathcal{B}_t$, we define $\mathcal{B}^p_t$ as collection of batches where any $B \in \mathcal{B}^p_t$ contains agents which are topologically connected in pessimistic set and $S^{p,B}_t := \cup_{i \in B} S^{p,i}_t$.

**Safe expansion**. Safe expansion is the sub-routine inspired by GOOSE for goal-oriented exploration of the safe set that we use to learn about the feasibility of sampling targets. It uses a heuristic $h$ to assign priority scores $p$ to points that are optimistically but not pessimistically safe. Those determine locations whose feasibility is relevant to learn that of the sampling targets ( Line 2 of Algorithm 3). A simple and effective choice for the heuristic is the inverse of the distance to the targets. Then, it identifies safe locations where the constraint is not yet known $\epsilon_q$-accurately (Line 3). Among them, it determines the $\alpha$-immediate expanders, i.e., those that could potentially add locations with priority $\alpha$ to the pessimistic set, $G^{\epsilon_q}_t(\alpha) = \{v \in W^{\epsilon_q}_t | \exists z \in A_t(\alpha) : u^q_t(v) - L_q d(v, z) \geq 0\}$. In Line 4, it selects the non-empty $\alpha$-expander set with the highest priority. In Line 6 - 7, the agent evaluates the constraint at the location with the highest uncertainty in this set (see [12] for details).

**SAFEMAC convergence**. The *optimistic coverage* phase switches to *optimistic exploration* phase, when density uncertainty under the disks is low ($w_t \leq \epsilon_\rho$). In the exploration, either the topological connection of the optimistic feasible set changes or will classify the uncertain region as pessimistically safe. In the former case, SAFEMAC will recompute a new coverage location and switch to the coverage phase. Alternatively, if the uncertain region is pessimistically safe, SAFEMAC has converged since the density uncertainty in the exploration phase is already low. The phases show an interesting dynamics; SAFEMAC continuously iterates between the *optimistic exploration* and the *optimistic coverage* phase until we know about the feasibility of the disk and their uncertainty is low. In the worst case, SAFEMAC might explore the entire environment. In this case the sample complexity will be similar to a two-stage algorithm, where we explore the whole domain and then optimize coverage in the resulting known environment. However, in practice, SAFEMAC is much better than this worst case.

# 5 Analysis

We now analyze MACOPT's convergence and SAFEMAC's optimality and safety properties.

**MACOPT**. To measure the progress of MACOPT, we study its regret, i.e., the difference between its solution and the one we could find if we knew the true density. Since control coverage consists in maximizing a monotone submodular function, we cannot efficiently compute the true optimum even for known densities. However, we can efficiently find a solution that is at least $(1 - 1/e)$ within the optimum. Thus, we quantify performance using the following notion of cumulative regret,

$$Reg_{act}(T) = \left(1 - \frac{1}{e}\right) \sum_{t=1}^{T} F(X_\star; \rho, V) - \sum_{t=1}^{T} F(X_t; \rho, V), \tag{5}$$

where $F(X_\star; \rho, V)$ is the optimal coverage. We now state one of our main results, which guarantees that the cumulative regret of MACOPT grows sublinearly in time (proof in Appendix D).

**Theorem 1.** *Let $\delta \in (0,1)$, $\beta_t^{\rho\,1/2} = B_\rho + 4\sigma_\rho \sqrt{\gamma_{Nt}^\rho + \ln(1/\delta)}$ and $C_D = \max_{x^i \in V} |D^i|/|V| \leq 1$. With probability at least $1 - \delta$, MACOPT's regret defined in Eq. (5) is bounded by $\mathcal{O}(\sqrt{T\beta_T^\rho \gamma_{NT}^\rho})$,*

$$Pr\left\{ Reg_{act}(T) \leq \sqrt{\frac{8C_D NT\beta_T^\rho \gamma_{NT}^\rho}{\log(1 + N\sigma_\rho^{-2})}} \right\} \geq 1 - \delta. \tag{6}$$

The proof of 1 builds on two key ideas. First, we exploit the conditional linearity of the submodular objective to bound the cumulative regret defined in Eq. (5) with a sum of per agent regrets. Secondly, we bound the per agent regret with the information capacity $\gamma_{NT}^\rho$, a quantity that measures the largest reduction in uncertainty about the density that can be obtained from $NT$ noisy evaluations of it. Since $\gamma_{NT}^\rho$ [21] grows sublinearly with $T$ for commonly used kernels, so does MACOPT's regret in Eq. (6). The immediate corollary of the above theorem, when the MACOPT stopping criteria is reached (Line 3 of Algorithm 2) guarantees a near optimal solution up to $\epsilon_\rho$ precision.

**Corollary 1.** *Let $t_\rho^\star$ be the smallest integer, such that $\frac{t_\rho^\star}{\beta_{t_\rho^\star}\gamma_{Nt_\rho^\star}} \geq \frac{8C_D^2 N^2}{\log(1+N\sigma^{-2})\epsilon_\rho^2}$, then there exists a $t < t_\rho^\star$ such that w.h.p, MACOPT terminates and achieves, $F(X_t; \rho, V) \geq (1 - \frac{1}{e})F(X_\star; \rho, V) - \epsilon_\rho$.*

**SAFEMAC**. This section presents our main result for safety-constrained multi-agent coverage control. In particular, Theorem 2 (proof in Appendix E) guarantees that SAFEMAC safely achieves near-optimal safe coverage in finite time.

**Theorem 2.** *Let $\delta \in (0,1)$, $\epsilon_\rho \geq 0$, $\|\rho\|_{k^\rho} \leq B_\rho$, $\beta_t^{\rho\,1/2} = B_\rho + 4\sigma_\rho \sqrt{\gamma_{Nt}^\rho + 1 + \ln(1/\delta)}$, $\gamma_{Nt}^\rho$ denote the information capacity associated with the kernel $k^\rho$. Let $q(\cdot)$ be $L_q$-Lipschitz continuous and $\epsilon_q, \beta_t^q, \gamma_{Nt}^q$ be defined analogously. Given $X_0 \neq \emptyset$, $q(x_0^i) \geq 0$ for all $i \in [N]$. Then, for any heuristic $h_t : V \to \mathbb{R}$, with probability at least $1 - \delta$, we have $q(x) \geq 0$, for any $x$ along the state trajectory pursued by any agent in SAFEMAC. Moreover, let $t_\rho^\star$ be the smallest integer such that $\frac{t_\rho^\star}{\beta_{t_\rho^\star}\gamma_{Nt_\rho^\star}} \geq \frac{8C_D^2 N^2}{\log(1+N\sigma^{-2})\epsilon_\rho^2}$, with $C_D = \max_{x^i \in V} \frac{|D^i|}{|V|} \leq 1$ and let $t_q^\star$ be the smallest integer such that $\frac{t_q^\star}{\beta_{t_q^\star}\gamma_{Nt_q^\star}} \geq \frac{C|\bar{R}_0(X_0)|}{\epsilon_q^2}$, with $C = 8/\log(1 + \sigma_q^{-2})$ then, there exists $t \leq t_q^\star + t_\rho^\star$, such that with probability at least $1 - \delta$,*

$$\sum_{B \in \mathcal{B}_t} F(X_t^B; \rho, \bar{R}_0(X_0^B)) \geq \left(1 - \frac{1}{e}\right) \sum_{B \in \mathcal{B}} F(X_\star^B; \rho, \bar{R}_{\epsilon_q}(X_0^B)) - \epsilon_\rho. \tag{7}$$

The theoretical analysis has two components: (*i*) we show SAFEMAC's coverage is near-optimal at convergence (Lemma 10), and (*ii*) we prove it converges in finite time. Since SAFEMAC learns the constraint *and* the density, we must bound the sample complexity for both to prove (*ii*). For the constraint, we extend the results for single-agent GOOSE to our multi-agent setting (Appendix F). For the density, we use results from Theorem 1 to show that, within a coverage phase, the cumulative regret is sublinear. Next, we use additivity of the information gain (Lemma 13) between any pair of coverage phases to bound the sample complexity of density for the subsequent coverage phases. Combining these results, we obtain Theorem 2.

**Intermediate recommendation**. Theorem 2 guarantees that SAFEMAC converges to a safe and near-optimal solution. Can it also make sensible recommendations before the stopping criteria are met?

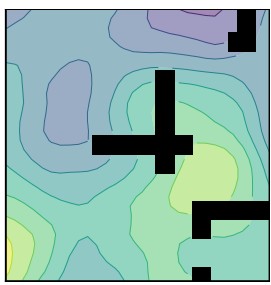

(a) Obstacles environment

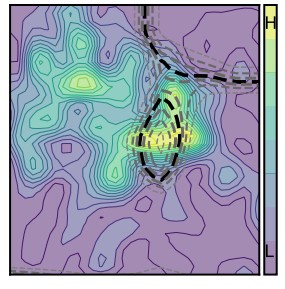

(b) Gorilla nest environment

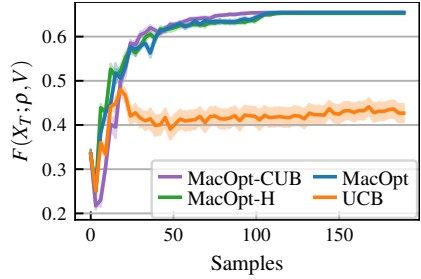

(c) Coverage on gorilla nests (Sunny day)

Figure 3: The contours in: a) show the synthetic density and the obstacles marked by the black blocks, b) show the Gorilla nests distribution with weather constraints marked by the black dashed line, and its contours with grey dashed line. c) Compares MACOPT with UCB in the safe gorilla environment. MACOPT does a more principled exploration of the coverage and does not stick to a local minimum.

Ideally, such recommendations should (*i*) be safely reachable and (*ii*) ensure a minimum coverage. To satisfy (*i*), they should be in the pessimistic safe set, $S_t^p$. To satisfy (*ii*), their coverage should be computed according to $F(\cdot; l_{t-1}^\rho, S_t^p)$, i.e., assuming a worst-case density, $l_{t-1}^\rho$, and a worst-case feasible set, $S_t^p$. If the greedy recommendation $X_t$ is in $S_t^p$, we can recommend it at intermediate steps. However, this is not always the case and we need an alternative. To this end, we compute $X_t^{l,B}$, i.e., the greedy solution w.r.t. the worst-case objective, $F(\cdot; l_{t-1}^\rho, S_t^{p,B}) \, \forall B \in \mathcal{B}_t^p$. At any time $T$, SAFEMAC recommends the best of either strategy up to time $T$ according to the worst-case objective. In Appendix E.1, we show that such recommendation is also near optimal at convergence.

## 6 Experiments

This section compares MACOPT and SAFEMAC to existing methods (or their extensions) on synthetic and real-world problems. We validate our theoretical claims and observe their superiority. We set $\beta^q = 3$ and $\beta^\rho = 3$ for all $t \geq 1$, it ensures safety as well as efficient exploration in practice [12]. Experiment details and extended empirical analysis are in Appendix G.

**Environments**. We perform our experiments with $N = 3$ agents in a $30 \times 30$ grid world where states are evenly spaced over $[0, 3]^2$. Each agent's disk is defined as the region an agent can reach in $r = 5$ steps in the defined grid. We normalize coverage with a maximum value $\sum_{v \in \bar{R}_0(X_0)} \rho(v)/|V|$. Below, we present the 3 environments we consider.

i) In *synthetic data*, both the density $\rho$ and the constrain $q$ are sampled from a GP with zero mean and Matérn Kernel with $\nu = 2.5$, scale $\sigma_k = 1$, and lengthscale $l = 2$. The observations are perturbed by i.i.d noise $\mathcal{N}(0, 10^{-3})$. ii) In *obstacles*, we sample maps with several block-shaped obstacles (Fig. 3a) and we aim to maximize coverage while avoiding dangerous collisions. At $v$, each agent senses the distance to the nearest obstacle $d_m(v)$, which could be given by sensors such as 1D-Lidars. We use $q'(v) = 1/(1 + \exp(-1.5 d_m(v)))$, to map the distance between $[0, 3]$ and saturate the constraint value for large distances, and we set $q(v) = q'(v) - 0.5$ to avoid collisions. The density is sampled from the same GP as the synthetic case. iii) In *gorilla nest*, we simulate a bio-diversity monitoring task, where we aim to cover areas with high density of gorilla nests with a quadrotor in the Kagwene Gorilla Sanctuary (Fig. 3b) . Regions affected by adverse weather (e.g. rain and storms) are unsafe for the drone due to higher chances of crashes and should be avoided. As a proxy for bad weather, we use the cloud coverage data over the KGS from OpenWeather [22]. The nest density is obtained by fitting a smooth rate function [23] over Gorilla nest counts [24].

**MACOPT**. We compare MACOPT to UCB, a baseline that skips the uncertainty sampling step from Section 4.1 and obtains measurements at the centers of the GREEDY sensing regions. We further develop two sample-efficient extensions of MACOPT: i) Correlated upper bound (CUB), a variant of MACOPT that constructs tighter upper confidence bound of the coverage function utilizing the covariance of density, instead of using the sum of density UCB. ii) Hallucinated uncertainty sampling (H), a variant of MACOPT that samples at the most informative location for each agent $i$, after hallucinating sampling locations of $\{1, \ldots i - 1\}$ agents. Please see Appendix D.1 for theoretical analysis. Fig. 3c shows a comparison in the *gorilla* environment on a day of good weather, i.e. when all locations are safe. Here, UCB gets stuck in a local optimum as it does not reduce the uncertainty

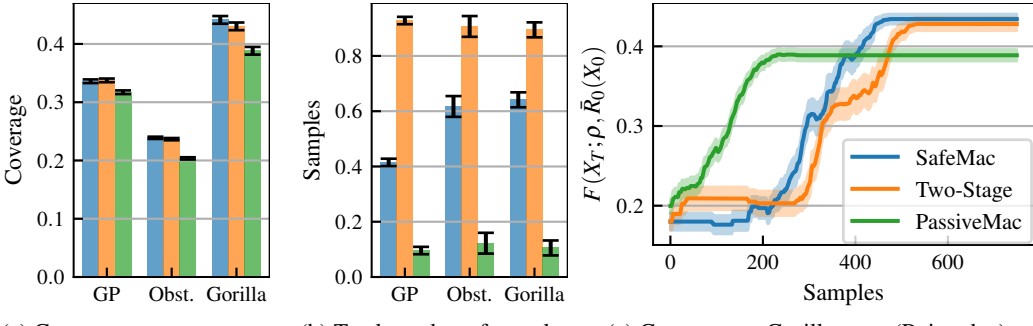

| (a) Coverage at convergence | (b) Total number of samples | (c) Coverage on Gorilla nests (Rainy day) |
|---|---|---|

Figure 4: Comparison of SAFEMAC with PASSIVEMAC and Two-Stage in all environments at convergence (a) and (b) and during optimization for the gorilla environment in (c). SAFEMAC trades-off learning about density and constraints, such that it finds a solution comparable to Two-Stage more efficiently, whereas PASSIVEMAC gets stuck in a local optimum.

of the density, whereas MACOPT explores more and achieves a higher coverage value up to 25%. Moreover, variants of MACOPT account for correlation and condition on other agents' measurement locations, which results in achieving the same coverage but more efficiently.

**SAFEMAC**. We compare SAFEMAC with two baselines: *i)* a two-stage algorithm [25], that first fully explores the feasible region, and then uses MACOPT to maximize the coverage; *ii)* PASSIVEMAC, a baseline inspired by [26] that runs MACOPT in the pessimistic set and passively measures the constraint in the process. Figs. 4a and 4b show the coverage at convergence and the number of samples to converge for SAFEMAC and the two baselines across all the environments. The results are averaged over 50 instances produced using different seeds and samples for every environment. In Fig. 4b, the y-axis is normalized with the maximum number of samples in the instance and then averaged over all instances. PASSIVEMAC converges quickly but gets stuck in a local optimum as it does not actively explore the constraint. SAFEMAC and Two-Stage converge to much higher coverage values. However, SAFEMAC is up to 50% more sample efficient thanks to its goal-oriented exploration. Fig. 4c shows the coverage value of the intermediate safe recommendations (Section 5) in the *gorilla* environment as a function of the number of samples. It confirms the previous results: SAFEMAC finds solutions comparable to Two-Stage more efficiently and PASSIVEMAC gets stuck in a local optimum.

**Scalability**. SAFEMAC utilizes the GREEDY algorithm, which is linear in the number of nodes (domain size). In each iteration, SAFEMAC computes a greedy solution $N$ times (one for each agent), which makes it linear in the number of agents. We model density and constraint using GP, which scales cubically with the number of samples. To demonstrate scalability in practice, we conducted experiments with $N = 3, 6, 10, 15$ agents each with domain length of $30, 40, 50$ and $60$ in Appendix G.1

## 7 Related work

Our work relates to multiple fields. We highlight the most relevant connections, referencing surveys where possible; an exhaustive overview is beyond the scope of this paper.

**Bayesian optimization**. In BO, an agent sequentially evaluates a noisy objective, seeking to maximize it [27]. In contrast, the quantity we measure *differs* from our objective. Partial monitoring [28] addresses such issues in an abstract setting [20, 29]. We exploit special structure in our problem. In coverage control with unknown density, this challenge is often addressed by learning the density uniformly over the domain [30, 31]. In contrast, MACOPT learns the density only at promising locations.

**Coverage control**. MAC with known densities is a well-studied NP hard [32] problem. Many algorithms use efficient heuristics to converge quickly to a local optimum. One popular strategy is Lloyd's algorithm [33], which has been studied in different settings, e.g., with known densities [34, 35], *a-priori* unknown densities [31, 36–38], using graph neural networks [39], taking into account agent's dynamics and constraints [40], or in case of non-identical robots [41]. These methods apply to continuous state and action spaces and show convergence to local optima, but lack optimality guarantees [30, 31, 40] and sample complexity bounds. Moreover, their extension to non-convex, disconnected domains is not trivial [42]. Coverage control is also studied in the episodic setting to learn the unknown policy or the environment using deep RL methods [43, 44].

**Submodular optimization**. Submodular functions are ubiquitous in machine learning [45] as they can be efficiently approximately maximized under different kinds of constraints [46]. For example, the GREEDY algorithm can be used in case of cardinality constraints [13] to maximize quantities like mutual information [47] or weighted coverage functions [15]. Online submodular maximization aims at optimizing unknown submodular functions from noisy measurements [48]. It has multiple applications, including optimization of numerical solvers [49], information gathering [50] and crowd-sourced image collection summarization [51]. Particularly related to ours is the work in [52], which proposes an algorithm for contextual news recommendation for linear user preferences with strong regret guarantees. In contrast to that setting, we consider dynamic agents, safety constraints and partial feedback.

**Safety**. Depending on the safety formulation and the assumptions, many algorithms have been proposed for safe learning in dynamical systems, e.g., based on model predictive control [53], curriculum learning [54], Lyapunov functions [55, 56], reachability [57], CMDPs [58], behavioral system theory [59], and more [60–62]. Here, we focus on the setting that is most closely related to ours, i.e., one with unknown but sufficiently regular instantaneous constraints that must be satisfied at all times. For stateless problems, e.g. BO, [26, 63] propose algorithms with safety and optimality guarantees with different exploration strategies. For stateful problems, [64] studies the pure exploration case, while [25] extends the two-stage approach from [63]. These approaches may be sample inefficient as they may explore the constraint in regions irrelevant for the objective. GOOSE [12] addresses this problem for both the stateful and stateless setting. The only work in this context that addresses multi-agent problems is [65]. However, their objective differs from ours, and they do not establish safety guarantees.

## 8    Conclusion

We present two novel algorithms for multi-agent coverage control in unconstrained (MACOPT) and safety critical environments (SAFEMAC). We show MACOPT achieves sublinear cumulative regret, despite the challenge of partial observability. Moreover, we prove SAFEMAC achieves near optimal coverage in finite time while navigating safely. We demonstrate the superiority of our algorithms in terms of sample efficiency and coverage in real-world applications such as safe biodiversity monitoring.

Currently, our algorithms choose informative targets but do not plan informative trajectories, which is crucial in robotics. We aim to address this in future work. Finally, while in many real-world applications the density and the constraints are as regular as assumed here, in some they are not. In these cases, our optimality and safety guarantees would not apply.

## Acknowledgements

Manish Prajapat is supported by an ETH AI Center doctoral fellowship. Matteo Turchetta is supported by the Swiss National Science Foundation under NCCR Automation, grant agreement 51NF40 180545. We would like to thank Pawel Czyz for insightful discussions.

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
