# OpenReview forum: "Near-Optimal Multi-Agent Learning for Safe Coverage Control"
_NeurIPS.cc/2022/Conference — NeurIPS 2022 Accept_

### Official Review · Reviewer_9Vst · 2022-07-11

**Rating:** 5
**Confidence:** 5
**Soundness:** 4 excellent
**Presentation:** 3 good
**Contribution:** 3 good

**Summary:**

The paper presents an algorithm for safe coverage and exploration with multiple robots.  Particularly, the paper focuses on an active information gathering problem where multiple mobile robots choose how to move around to explore the environment and maximize its coverage.  Coverage is modelled through a spatial Gaussian process density function.  Exploration is necessary since the density is unknown a priori.  Safety constraints are also considered (the robots cannot access all locations in the environment), which are also assumed unknown a priori.  The constraints are also modeled via Gaussian Processes.  The paper provides an algorithm that guarantees (i) near-optimal coverage and (ii) satisfaction of the safety constraints. The algorithm is evaluated on a synthetic and two real world applications: safe biodiversity monitoring and obstacle avoidance.

**Questions:**

1.  I would elaborate on how the chosen kernel functions are known a priori.
2.  I would elaborate on how dynamics constraints are considered.  For example, can robots move arbitrarily within the safety sets?
3.  I would elaborate on how scalable the proposed method is.  What is its computational complexity? Up to how many robots can it scale?


**Limitations:**

1.  The paper discusses in the conclusion that the robots currently are allowed to jump from any point in the safety set to any other; the conclusion states that future work will address the problem of planning feasible trajectories.
2.  I would elaborate on the limitations of modeling the density and safety constraints as Gaussian Processes.
3.  I would elaborate on how scalable the proposed method is.


**Strengths And Weaknesses:**

Strengths
(+)  Proposed algorithm guarantees sublinear regret and safety
(+)  Algorithm’s effectiveness is illustrated in simulations

Weaknesses
(-)  Density and constraints have a known structure, being modeled as Gaussian Processes given kernel functions.  I would elaborate on how the chosen kernel functions are known a priori.
(-)  The robots are allowed to jump from any point in the safety set to any other.
(-)  The experiments consider robot teams of only 3 robots.

---

> ### Author Response · Authors · 2022-07-31
> **Response to reviewer 9Vst**
>
> We thank reviewer 9Vst for the time and the insightful feedback.
>
> **Known kernel:** Kernel functions express our prior belief about the unknown density and constraint. If domain knowledge is available, it can be readily incorporated in them. If this is not the case, one can use conservative kernels that capture large classes of functions (e.g. Matern with low \nu and small lengthscales). There is a trade-off: the more conservative the kernel the higher the likelihood that the function we are modelling is in the corresponding RKHS but the less efficient the exploration. In the literature [11,22,23,57,58], it is common to assume that domain knowledge is used to select a kernel that strikes the right trade-off. Thanks for bringing this up, we will elaborate more on this in the final version of the paper.
>
> **Dynamics constraints and Path planning:** Please see the common response at the top.
>
> **Scaling:** Please see the common response at the top.
>
> **Modelling using GP:** If the prior is wrong (refer to known kernel above), modelling of constraints and density might not be error-free. However, sample efficiency, uncertainty quantification, and analysis still make GP a very suitable choice.

---

### Official Review · Reviewer_p24T · 2022-07-11

**Rating:** 6
**Confidence:** 4
**Soundness:** 4 excellent
**Presentation:** 3 good
**Contribution:** 2 fair

**Summary:**

This paper looks into multi-agent coverage control (MAC) problems, in which multiple agents coordinate and navigate the environment to maximize the coverage of some densities. It aims at learning the density to solve the problem while ensuring the safety of the agents. It formally defines the safety-constrained multi-agent coverage control problem. The MAC task is modeled as a conditionally linear coverage function that possesses monotonocity and submodularity. A single-agent safe exploration algorithm named GOOSE is introduced. Using GOOSE as the intuition, the authors propose MACOPT, an unconstrained multi-agent coverage control algorithm, and SAFEMAC, a safety-constrained multi-agent coverage control algorithm that is created by extending GOOSE to multi-agent cases and combining it with MACOPT. Then they prove the convergence of MACOPT and the optimality and safety properties of SAFEMAC. Finally, the paper discusses how MACOPT and SAFEMAC are superior compared to existing methods by doing experiments in synthetic and real-world applications. It shows that SAFEMAC obtains better solutions than algorithms that do not actively explore the feasible region and that it has higher sample efficiency than competing near-optimal safe algorithms.

**Questions:**

In the Introduction section, the authors mentioned that deploying coverage control solutions in the real world presents many challenges and no prior work addressed them jointly. One of the challenges is that multi-agent coverage control with known densities is an NP-hard problem. Has the paper tried to address this specific problem?

In the Conclusion section, the authors noted that although in many real-world applications the density and the constraints are as regular as assumed in the paper, in some they are not. Is there a specific example in which the algorithms do not apply well?

**Limitations:**

The authors specified that the limitation of the paper is that the proposed algorithms choose informative targets without planning informative trajectories, which can be crucial in the research of robotics. Besides, in some real-world applications, the density and the constraints may not be the same as assumed in the paper. In these cases, the algorithms no longer have optimality and safety guarantees. It is not likely that this paper will cause any potential negative social impact.

**Strengths And Weaknesses:**

This paper introduces a single-agent safe exploration algorithm (GOOSE) as the background, extends it to a multi-agent version, and proposes two algorithms for unconstrained and safety-constrained multi-agent coverage control problems. It starts from a well-known method and then presents novel approaches inspired by it, which demonstrates good originality.

The submission is written in a very solid style. The problem statement and definitions are formal, and the pseudocodes for MACOPT and SAFEMAC are displayed in detail. The paper analyzes MACOPT ’s convergence and SAFEMAC ’s optimality and safety properties using theorems and mathematical proofs. The statements are accurately written and rigorously proved (the full derivation is included in the appendix). The structure of the main paper is clear and organized. The authors demonstrate that multi-agent coverage control tasks are a class of difficult problems, especially when safety needs to be guaranteed, and their new methods provably address the tasks more efficiently than previous works.

On the other hand, the description of the figures should be improved. In the caption of Figure 1, it says "Agent 1 covers $D^1$ (green), 2 covers $D^{2-}$ (orange) and 3 covers $D^{3-}$ (yellow)." However, I cannot see where $D^1$, $D^{2-}$ and $D^{3-}$ are labeled in Figure 1(a). It also says "In b) ... in the optimistic set", but this set is not marked and its color is not specified in Figure 1(b) either.

Another side note is about the grammar problem. For instance,  in line 108, "... a the positive definite kernel matrix ..." should be "... the positive definite kernel matrix ..." and in line 265, "... such 1D-Lidars." should be "... such as 1D-Lidars."

---

> ### Author Response · Authors · 2022-07-31
> **Response to Reviewer p24T**
>
> We are thankful to the reviewer p24T for their time and constructive feedback. We are glad that the reviewer recognises our formulation, performance guarantees, and algorithms to be novel and written in a solid style.
>
> **NP-hard problem:** We need to approximately solve the NP-hard problem of maximizing a monotone submodular function under cardinality constraints as a sub-routine at each iteration. To this end, we use the greedy algorithm that is guaranteed to recover a solution within a factor 1-1/e from the optimal one [12], which is provably the best approximation we can obtain in polynomial time [46]. Therefore, we do address this problem directly by leveraging existing results.
>
> **Function regularity:** Our method relies on the idea that safety in one location is predictive of the safety of nearby ones. Therefore, SafeMac would struggle in environments where agents may go abruptly from a very safe region to a very unsafe one. E.g., if we define safety in terms of the steepness of the terrain explored by a rover (see  [11, 22, 58]) the agents may go from 0 (safe) to very high steepness (unsafe) if there is a cliff. In such cases, the agent may wrongly predict the safety of nearby locations.
>
> Thank you very much for pointing us to caption clarity and grammatical errors, we have included all of them in the paper.

---

### Official Review · Reviewer_EbSJ · 2022-07-16

**Rating:** 6
**Confidence:** 3
**Soundness:** 3 good
**Presentation:** 3 good
**Contribution:** 2 fair

**Summary:**

The paper proposes a multi-agent coverage algorithm for a 2D environment with a priori unknown unsafe regions and noise density estimates that is guaranteed to be within a constant factor away from the optimal solution under certain assumptions (e.g. well behaved density and safety functions).

**Questions:**

Would the problem under the assumption of conditionally linear, monotone and submodular functions still be NP hard?

I am not sure if Algorithms 1 and 4 are fully correct. In Lines 3 and 5, respectively, initially when t=0, they probe w_0, which is not initialised. Should this be w_1?

Corollary 1 seems to suggest that choosing t_p^* = 0 always works. I think it should be >= not <=.

In Section 5 (Analysis), you state "Since control coverage consists in maximizing a monotone submodular function, we cannot compute the true optimum even for known densities". Why is this the case (in general, that is, non-greedy approaches)? Could one not simply enumerate through all solutions?

How would the method scale with the number of agents (K beyond 3)?

**Strengths And Weaknesses:**

The strength of the paper is a novel formulation of the exploration vs exploitation problem that takes into account agent safety in a multi-agent setting as well as the performance guarantees obtained. Although the algorithms and analysis built on previous work (e.g. [11,16,23]), I thought the work to be sufficiently novel.

The performance guarantee (Theorem 2) states that by doing enough iterations (i.e. sampling as a function of epsilon and delta), one can guarantee that the solution quality gets arbitrarily close (i.e. epsilon) to within a constant factor of 1-1/e of the optimum [10] with probability 1 - delta. Hence, the claim in the abstract of "near optimal coverage in finite time" seems overstated. In fact, when running the algorithm for a finite time, with probability delta > 0 one may end up worse than within epsilon from the constant factor approximation. Moreover, even if running the algorithm indefinitely, one can end up 1/e which corresponds to about 37% away from the optimum, which is good, but may not be considered as "near-optimal" by everyone.

The main weakness of the paper is that:
(i) path planning of individual agents is not considered (samples are taken from regions of sufficiently safe cells, hence, ignoring the time and risks that accumulate for an agent as it travels through this region);
(ii) it is not clear to what extent the proposed methods scale as only K=3 agents are considered and the environment is relatively small considering the intended applications (30 x 30 cells) albeit it is not a toy problem. Given much prior work exists for K=1, I would have appreciated an evaluation of the methods (that specifically address the multi-agent scenario) in more complex settings.

The paper claims that constraints prevent "agents from monitoring from unsafe locations" and agents "may be unable to safely reach a disconnected safe area" starting from their initial locations. The problem statement assumes that agents can only monitor locations that they can (in principle) reach, as they have to be part of the subset of the largest "safely reachable region" (see equation 2). This prevents an agent from monitoring say the other side of a river if the river cannot be safely crossed, irrespective of their sensing range. Although reasonable (to not overly complicate the problem statement), this assumption should be explicitly acknowledged, as the optimality guarantees are not correct otherwise.

The problem statement is challenging to follow, given there are a number of inter-connected aspects. Up to equation (2) it is rather straightforward, however, given uncertainties are introduced regarding the density and safety it is less clear thereafter what is actually the objective. For example, if the objective was to obtain the best possible result over a fixed duration - which may be realistic objective in practice - than the methods would not necessarily deliver, as the solutions presented do not seem to be anytime algorithms. Instead, the objective seems to be provide a solutions however long it takes (see also minor suggestions below).

A few assumptions may make it harder to implement the method in practice, including the need for a centralised optimisation and synchronous measurements as well as the functions needed to be continuous (if they are not, the advantage of the method may be reduced). Also, looking at Figure 3(c), it seems there is a certain window (in terms of number of samples) where SafeMac excels whereas if there are fewer samples the method is clearly outperformed by PassiveMac, and as the number of samples approach 800 (which is almost 900 = one sample per cell) I wonder whether other solutions exists that perform equally well or better.

I am not sure if Algorithms 1 and 4 are fully correct. In Lines 3 and 5, respectively, initially when t=0, they probe w_0, which is not initialised. Should this be w_1?

Corollary 1 seems to suggest that choosing t_p^* = 0 always works. I think it should be >= not <=.

In Section 5 (Analysis), you state "Since control coverage consists in maximizing a monotone submodular function, we cannot compute the true optimum even for known densities". Why is this the case (in general, that is, non-greedy approaches)? Could one not simply enumerate through all solutions?

I could not find a definition of L_q (input of Algorithm 4).

Minor comments:
- When talking about approximations of the feasible sets, are the max and min operators over x?
- You first refer to Algorithm 2, then to Line 4 in Algorithm 1, and then you write "Now, we introduce MACOPT (Algorithm 1)", which is not ideal.
- Where is [11] published?
- In Theorem 2, you start with "Let delta be element of (0,1)", but the same could be said for epsilon, as you chose to use 2 variables instead of one.
- "a expander"

---

> ### Author Response · Authors · 2022-07-31
> **Response to Reviewer EbSJ**
>
> We thank the reviewer EbSJ for the detailed review and the constructive feedback. We are glad that the reviewer acknowledges the novelty of the work.
>
> Thank you very much for letting us know about the typos and clarifications, we have included all of them in the paper.  Please see our responses to the major concerns below:
>
> **Strength and weakness**
>
> **Near-Optimal in finite time:** In combinatorial optimization, constant factor approximation guarantees and those matching lower bounds (as is the case in our paper) in particular are referred to as near-optimal. High-probability guarantees are standard in the literature when working with GPs (GP-UCB[17], Kernelized Bandit [16]) due to the unbounded support of Gaussians.
>
> **Safe path planning and scaling of the algorithms:**  Please see the common response at the top.
>
> **Objective of the problem:** Both fixed-budget (the problem suggested by the reviewer) and fixed-confidence (the problem we address in the paper) are widely studied settings in the bandit literature and both have relevant applications (see “Best Arm Identification: A Unified Approach to Fixed Budget and Fixed Confidence” by Gabillon et al.2012). We believe that a fixed-budget algorithm for this problem could be an interesting research direction and we will clarify this in the paper.
>
> **Why PassiveMac perform better initially?:** It is intuitive that PassiveMac performs better than SafeMac initially, in many cases. PassiveMac focuses on high-objective values and converges quickly to a local optimum as it does not actively explore the constraints. In contrast, SafeMac focuses on samples that enlarge the safe set but may not have a high value initially. However, this exploration of the constraints, independent of the objective value pays off in the long run as SafeMac explores the safe set to discover a better solution.
>
> **Number of samples: In fig 3c,** The X-axis represents the sum of both the density and the constraint samples. Nearly 800 corresponds to the total samples in the longest episode of the two-stage algorithm, whereas SafeMac converges early (<600 samples, Fig 3b show the relative samples). Moreover, the number of samples required is determined by the shape of the constraint function, the required accuracy, and the noise.
>
> **Questions:**
> - Yes, under the assumption of conditionally linear, monotone and submodular function, the problem is still NP-Hard.
> - The brute force approach of enumerating all the solutions is intractable for all but trivial problems. We meant under reasonable complexity-theoretic assumptions, there is no polynomial time algorithm that can give the solution with a known density.
>
> [11] is published in NeurIPS 2019; apologies that the .bib of the paper didn’t include it.

---

> > ### Comment · Reviewer_EbSJ · 2022-08-08
> > **Unaddressed questions**
> >
> > I can see some revisions have been made to the documents (e.g. new Figure 6 in the Appendix), however, it seems the following of my questions were not addressed.
> >
> > 1) I am not sure if Algorithms 1 and 4 are fully correct. In Lines 3 and 5, respectively, initially when t=0, they probe w_0, which is not initialised. Should this be w_1?
> >
> > 2) Corollary 1 seems to suggest that choosing t_p^* = 0 always works. I think it should be >= not <=.

---

> > > ### Author Response · Authors · 2022-08-08
> > > **Response to Reviewer EbSJ**
> > >
> > > Thanks for your reply. Apologies that we didn’t answer individual typos/clarification, we wanted to keep the response short for the reviewer time.
> > >
> > > 1. The algorithms shall start with t=1 instead of t=0. All the notations are as per that.
> > > 2. Yes, you are right, the sample complexity bound in corollary 1 should be >=. (similar to theorem 2, line 232)
> > >
> > > We will address all the typos and the required clarifications in the final version.

---

### Official Review · Reviewer_rxa3 · 2022-07-22

**Rating:** 5
**Confidence:** 1
**Soundness:** 2 fair
**Presentation:** 2 fair
**Contribution:** 2 fair

**Summary:**

Two algorithms to deal with a map covering problem where there are unknown obstacles is proposed. The first algorithm is a single agent algorithm in which it needs to deal with the exploration-exploitation dillema because of the partial observability of the environment.
The second algorithm, extends the first algorithm by considering multi-agent settings in which the exploration as well as the safe coverage is a chanllenge. It is shown that the second algorithm obtain near optimal coveragw while guarantees the safety.

**Questions:**

Q0- Would your algorithm works OK with larger number of agents? For example [3] and [4] run problems with 20 and 10 agents, respectively. Is it possible for you to run a problem with a same or bigger size?
Similarly, [5] has a 4 agents problem and the analyzed environment is also pretty similar to yours. That also could be a good benchmark.

Q1- I have seen several RL and MARL algorithm to deal with this problem. Even though you propose a sublinear cumulative regret, but still you do not know if the RL based algorithm can achieve the same performance or not on a same problem. I think an interseted reader would like to see the comparisons of these two classes of algorithms.

[1] A reinforcement learning‐based approach for modeling and coverage of an unknown field using a team of autonomous ground vehicles
[2] Adepegba, Adekunle A., Suruz Miah, and Davide Spinello. "Multi-agent area coverage control using reinforcement learning." The Twenty-Ninth International Flairs Conference. 2016.
[3] Ye, Zhenhui, et al. "Multi-UAV Navigation for Partially Observable Communication Coverage by Graph Reinforcement Learning." IEEE Transactions on Mobile Computing (2022).
[4] Gosrich, Walker, et al. "Coverage Control in Multi-Robot Systems via Graph Neural Networks." 2022 International Conference on Robotics and Automation (ICRA). IEEE, 2022.
[5] Battocletti, Gianpietro, et al. "RL-based Path Planning for Autonomous Aerial Vehicles in Unknown Environments." AIAA AVIATION 2021 FORUM. 2021.
[6] Din, Ahmad, et al. "A deep reinforcement learning-based multi-agent area coverage control for smart agriculture." Computers and Electrical Engineering 101 (2022): 108089.
[7] Faryadi, Saba, and Javad Mohammadpour Velni. "A reinforcement learning‐based approach for modeling and coverage of an unknown field using a team of autonomous ground vehicles." International Journal of Intelligent Systems 36.2 (2021): 1069-1084.


**Limitations:**

Same as above

**Strengths And Weaknesses:**

Same as below

---

> ### Author Response · Authors · 2022-07-31
> **Response to the Reviewer rxa3**
>
> We would like to thank reviewer rxa3 for the valuable feedback.
>
> Please see our answers below:
>
> **Summary of the paper:** We would like to clarify that the first algorithm (MacOpt) is also a multi-agent one.
>
> **Scaling:** Please see the common response at the top.
>
> **Related literature:**  The coverage control literature using RL considers settings and applications that differ considerably from ours. The major differences are,
> - They consider an episodic setting whereas we consider a non-episodic one where agents learn a near-optimal solution in a single trajectory.
> - [2,3,4,6] do not consider constraints. [5] uses heuristics and tuning to satisfy constraints but it incurs violations during training. In contrast, SafeMac guarantees safety at all times.
> - [1,2,4] solve a different problem (using Lloyd’s algorithm, more in our related works). Moreover, almost all of them are developed for working with a different sensor observation model (e.g, patch observation of nearby cells) [1,3,4,5,6].
>
> The differences above make a suitable and fair comparison rather difficult, if not impossible. However, we will include several of the works mentioned here in the related works.

---

### Author Response · Authors · 2022-07-31
**Common Response**

We thank all the reviewers for their time and valuable feedback. Here, we address common questions raised by most reviewers.

**Scaling:** Our algorithm evaluates a greedy solution K times (one for each agent) at each iteration. Therefore, it is linear in the number of agents. Moreover, the greedy algorithm is linear in the number of cells (domain size). To demonstrate scalability in practice, we added experiments with 3, 6, 10 and 15 agents each with domain sizes 30x30, 40x40, 50x50 and 60x60 in appendix G.1 (Page 44, last page in the revised supplementary).

**Path Planning:**
- **Agent moving within the safe set:** We do *NOT* assume that agents can jump within the safe set. Instead, given two pessimistically safe nodes in the graph, our analysis guarantees that there is a path within the pessimistic safe and ergodic set connecting them. Therefore, we can always obtain a safe path using Dijkstra or A*. In reality, the robots would use a low-level controller (e.g., PID or MPC) to track this high-level path.
- **Safety @ EbSJ:** Since this path is contained in the safe set, the agents follow a safe trajectory, if it tracks it sufficiently well (which can be achieved with a well-designed low-level controller).
- **Dynamics constraints @ 9Vst:** The dynamics are modelled using a graph where nodes represent states and edges represent transitions. This representation is general and can embed dynamic constraints (e.g. a 4-wheeled vehicle could go forward, forward left, forward right but not just left or right).

Apologies if this was not clear in the paper, we will elaborate more on this in the final version of the paper.

---

> ### Comment · Reviewer_EbSJ · 2022-08-08
> **Scalability analysis**
>
> Figure 6 explores performance for a wider range of conditions (scalability). As in the original submission there appears to be a region where the proposed approach is most useful (for low to medium number of samples, PassiveMac is far better, for high number of samples, Two-Stage is about equal or not much worse. It may be sensible to use a same y-axis range per row. At the moment, the range is different depending on both row and column which makes it hard to compare the results.

---

> > ### Author Response · Authors · 2022-08-08
> > **Response for scaling**
> >
> > Thank you for your response. The main goal of the empirical study is
> >
> > >(\*) “to show that SafeMac finds a comparable solution to two-stage in a much sample-efficient way whereas PassiveMac quickly gets stuck in a local optimum.”
> >
> > With the scaling experiment, we show that (\*) also holds for a large number of agents and larger domains. This is clear from any plot in figure 6.
> >
> > Thanks for the feedback to keep the same y-axis range, it may not provide new information about (\*). However, it can be interesting to see the coverage trend with scaling (larger K and domain size). We will include it in the final version of the appendix.

---

> > > ### Author Response · Authors · 2022-08-09
> > > **Small clarification regarding PassiveMac**
> > >
> > > > As in the original submission there appears to be a region where the proposed approach is most useful (for low to medium number of samples, PassiveMac is far better), for high number of samples, Two-Stage is about equal or not much worse.
> > >
> > > For the reviewer comment above, we would like to add that,
> > >
> > > PassiveMac is a heuristic that does not have theoretical guarantees. It converges to a local optimum, which can be far from optimum as well depending on the environment (in particular based on how the unknown constraint function looks like), e.g. for the obstacle and the GP environment, in fig 5a,b (in the appendix, Page 42) PassiveMac curves are different.
> > >
> > > Both Two-stage and SafeMac have optimality and safety guarantees. However, the Two-stage algorithm requires more samples, as shown in fig 3b.

---

### Author Response · Authors · 2022-08-04
**Message to all the reviewers**

Dear Reviewers,

We thank all of you for your time and valuable feedback. We believe that we have addressed all your concerns. If you have further doubts, we would be happy to discuss them. Otherwise, please consider raising your scores.

Thanks,

---

### Meta-Review · Area_Chair_owzE · 2022-08-25

**Recommendation:** Accept
**Confidence:** Less certain

**Metareview:**

This paper presents a novel method for multi-agent coverage control over an unknown density and safety constraints. There is some concern about the level of significance of the approach but it is interesting and sound. There were also concerns about scalability and the use of GPs for density modeling but the authors have sufficiently addressed these in the response and updated paper. The paper would be strengthened by highlighting the contributions and more extensive experiments to show the benefits of the approach in different settings.

**Award:**

No

---

### Decision · Program_Chairs · 2022-09-14

Accept